# SAFE, a new therapeutic intervention for families of children with autism: a randomised controlled feasibility trial

Rebecca McKenzie ,[1] Rudi Dallos,[2] Jacqui Stedmon,[2] Helen Hancocks,[3] Patricia Jane Vickery,[3] Andy Barton,[4] Tara Vassallo,[1] Craig Myhill,[1] Jade Chynoweth,[5] Paul Ewings[6]

► Prepublication history and supplemental material for this paper is available online. To view these files, please visit the journal online (http://dx.doi.org/10.1136/bmjopen-2020-038411).

¹Institute of Education, University of Plymouth, Plymouth, UK
²Department of Clinical Psychology, University of Plymouth, Plymouth, UK
³Peninsula Clinical Trials Unit, University of Plymouth, Plymouth, UK
⁴South West Research Design Service, Plymouth, UK
⁵Medical Statistics, University of Plymouth, Plymouth, UK
⁶Research Office, Research Design Service, Taunton, UK

**Correspondence to**
Dr Rebecca McKenzie;
rebecca.mckenzie@plymouth.ac.uk

## ABSTRACT

**Objectives** To establish the feasibility of a definitive randomised controlled trial of Systemic Autism-related Family Enabling (SAFE), an intervention for families of children with autism.

**Design** A randomised, controlled, multicentred feasibility study.

**Setting** Participants were identified from three National Health Service (NHS) diagnosing centres in Plymouth and Cornwall and a community pathway.

**Participants** 34 families of a child with a diagnosis of autism severity level 1 or 2 between 3 and 16 years. Four families were lost to follow-up.

**Interventions** SAFE is a manualised five-session family therapy-based intervention delivered over 16 weeks and designed for families of children with autism. SAFE involves families attending five 3-hour sessions led by systemic practitioners.

**Primary and secondary outcome measures** The proposed primary outcome measure was the Systemic CORE 15 (SCORE-15). Proposed secondary outcome measures: Patient Health Questionnaire-Somatic Anxiety Depressive Symptoms, the Coding of Attachment-Related Parenting for use with children with Autism, the Child Behaviour Checklist (CBCL), the Reflective Functioning Questionnaire (RFQ) and the Caregiving Helplessness Questionnaire. Outcome measures were collected at baseline and 24 weeks post randomisation.

**Results** All primary caregivers retained in the study completed the SCORE-15 at both time points. 34 of the target of 36 families were recruited and 88% of families were retained. Training for therapists was effective. Feedback revealed willingness to undergo randomisation. There was 100% attendance at appropriate sessions for core family members. The SCORE-15 showed reduction in scores for families receiving SAFE compared with controls suggesting positive change. Qualitative data also revealed that families found the study acceptable and families receiving SAFE experienced positive change. Feedback indicated that the SCORE-15 should be retained as a primary measure in a future trial, but secondary measures should be reduced.

**Conclusions** This study indicates that a larger trial of SAFE is feasible. Findings suggest that SAFE can address current gaps in recommended care, can be confidently delivered by NHS staff and has potential as a beneficial treatment.

### Strengths and limitations of this study

► The study addressed a gap in the available research data, and produced important feasibility information to inform a fully powered randomised controlled trial.
► The study explored the feasibility of using measures of family function and a range of mental health measures.
► Quantitative feasibility data were complemented by qualitative focus groups and interviews.
► The study explored the feasibility of economic analysis measures in a population, which includes adults and their children with developmental disorders.
► The participants were recruited from two National Health Service Trusts in adjacent counties in the South West of England, leading to potential bias. A future randomised controlled trial will extend to centres across the UK including Scotland and Wales.

**Trial registration numbers** ISCTRN83964946 and IRAS213527.

## INTRODUCTION

More than 1% of the UK population has a diagnosis of autism.[1] Families of children with a diagnosis of autism present complex needs. Children with autism have impairments in social communication and restricted, repetitive behaviours and interests.[2] Autism is widely accepted to have a genetic component and the broader autism phenotype is disproportionally represented among family members.[3] Mental health problems are experienced by more than 70% of individuals with autism and more than 50% of their parents.[4 5] Parents of children with autism are more likely to be hospitalised for mental disorders than parents of typically developing children[6] and mothers of children with autism are reported to have higher unmet needs, more difficulties coping and lower satisfaction with

service interactions than mothers of children with other disabilities.[7]

As these families often exhibit psychological morbidities alongside autism in the children, costs of services to treat these problems are high.[8 9] Furthermore, untreated or unresponsive mental health problems impose societal costs making it hard for parents to interact effectively with services,[10] potentially worsening outcomes for children and exacerbating the substantial economic burden of autism.[9]

Explanations for high levels of affective disorders in these families include: stress associated with the condition of autism, genetic factors and intergenerational family dynamics. Parenting children with autism involves stresses associated with challenging behaviour, communicative difficulties, isolation and atypical attachment behaviour displayed by children.[11] In addition, these parents suffer from poorer quality of life, and higher levels of stress compared with other populations due to societal issues including stigma, unemployment and difficulty accessing support.[12 13] Parents of children with autism report that a consequent lack of psychological well-being exacerbates maladaptive behaviour in their children,[14] which is likely to result in unhelpful cycles of distress and hopelessness.

Previous research demonstrates that experience of trauma and abuse among women is associated with elevated risk of autism developing in their subsequent offspring.[15 16] Hence, it is possible that mothers of children with autism are more likely than the general population to be coping with previous traumatic events. Families of children with autism can experience positive family life, cope well with difficulties and enjoy good relationships with their children,[17] but they represent a high-risk group, for whom treatment is disjointed, costly and inadequate.[18 19]

A more joined-up approach is required, focusing on communication within families, strategies for coping with challenging behaviour and associated problems and building on strengths and relationships to alleviate mental health difficulties. The current study should be placed in the context of international calls for improved services and care for families of children with autism at country level,[20] alongside National Institute for Health and Care Excellence (NICE) guidelines and recommendations,[21 22] as well as developments regarding children's service provision proposed by the Monroe report.[23 24] Families of children with autism themselves highlight the importance of professionals working therapeutically with children and the wider family.[6]

Systemic Autism-related Family Enabling (SAFE) is a systemic family therapy approach designed to address autism-related needs including problem-solving, poor sociocommunication, mental health difficulties and challenging behaviour.[25] Systemic family therapy is a well-recognised, evidence-based psychotherapeutic approach.[26] Despite evidence that family therapies can provide benefits to children with autism and their parents[27 28] its efficacy for addressing challenges associated

with this condition has not been subject to a randomised controlled trial. In addition, existing research overwhelmingly shows that families of children with autism want interventions which make real improvements to their daily life and sense of well-being.[29 30]

## Patient and public involvement

This study grew from the articulated needs of over 90 families surveyed and interviewed through the Plymouth Autism Network, specifically the lack of support postdiagnosis and the need for support for the whole family. The main aim was to develop a family-orientated support package and conduct a feasibility study, definitive trial and ultimately implement an intervention to be offered to all families after diagnosis. To achieve this aim in a sustainable manner we included families as partners at every stage of design and application.

Prior to commencement of the trial a pilot study was conducted with families acting as consultants. From the pilot, we recruited interested families to form the SAFE Family Consultation Group. Their representative was a coapplicant on the initial bid and was employed as a research assistant. The consultation group were consulted at every stage of the trial including the initial application, developing, refining and administering the intervention, developing trial materials, recruitment, training and dissemination.

The family representative attended all trial management and research meetings and took part in all training. She disseminated to consultation group members and where appropriate, meetings with the wider group were held. The input from the families has been invaluable in engaging participants and ensuring their well-being.

Patient and public involvement (PPI) was an intrinsic part of this study and our continued positive engagement and partnership with families of children with autism is a strength. The research emerged from the difficulties articulated by families and we worked with them to develop solutions. We feel that this way of working made a substantial impact on the outcomes for the families receiving the intervention and the fact that the feedback from families was overwhelmingly positive. We also believe that PPI ensured a more ethical and thorough study which did not lose sight of the ultimate aim. The input of families throughout provided considerable motivation for the research team in delivering the study despite the challenges faced.

## Aims and objectives

The aim of the study was to establish the feasibility of a definitive randomised controlled trial of SAFE.

Our objectives were to:

1. Demonstrate ability to identify, recruit and randomise eligible families.
2. Verify that proposed outcome measures and follow-up were acceptable, and targets for loss to follow-up were achievable.
3. Assess adherence of families to the intervention.

4. Gather quantitative data on outcomes to inform the design (and sample size) of the future trial.
5. Adapt existing Resource Use Questionnaire (RUQ) and assess the feasibility of preference based instruments for this population to facilitate a future economic evaluation.
6. As part of the feasibility of process evaluation, collect data on the families' experience of SAFE and the study itself.
7. As part of the feasibility of process evaluation, ensure that proposed training arrangements were effective and scalable.
8. Provide operational experience to manage the future trial.

## METHODS AND ANALYSIS

The study comprised a randomised, controlled, multi-centred feasibility study including families of children with a diagnosis of autism.[25]

### Participants

Participants were identified from three National Health Service (NHS) diagnosing centres in Plymouth and Cornwall via a search of clinical records or recruitment directly after diagnosis. A community recruitment pathway was added after commencement of the study to alleviate the burden on clinical staff and increase participation. Families were recruited by posters in local community venues.

Eligibility was determined by diagnosing clinicians or from clinical records or diagnostic letter as well as discussion with the family. A member of the research team gained consent during the first home visit. Pilot data suggested that SAFE was most effective and accessible for children without severe symptoms or an intellectual impairment. Children with severe communication difficulties found it difficult to engage with some SAFE activities. For this feasibility study, therefore, our target population was families of children with autism severity level 1 or 2 with no intellectual impairment.

### Eligibility criteria

► Family included child with autism, aged 3–16 years.
► Diagnosis of Autism Spectrum Disorder (ASD) severity level 1 or 2 as documented in Diagnostic and Statistical Manual of Mental Disorders, Fifth Edition (DSM-5).[2]
► Diagnosed within 12 months of consenting to the study.
► If other diagnoses were present, ASD must be primary diagnosis.
► Families were willing to comply with study requirements.

### Exclusion criteria

► Child with ASD severity level 3 as documented in DSM-5.[2]
► Child with ASD and intellectual impairment.

► Serious concomitant illness in child or family, or other circumstances such that they were unable to comply with study requirements.
► Families who may have been a risk to safety of research staff.
► Insufficient English language, or capacity for parent/child to consent/assent to the study.

### The intervention

SAFE is a manualised intensive programme of systemic family therapy designed to treat maladaptive autistic symptoms and mental health-related difficulties encountered by families of children with autism (online supplemental appendix 1). SAFE lead therapists were qualified NHS systemic practitioners who received an additional 4-day training course in SAFE principles and delivery. Each lead therapist was accompanied by a support therapist trained in SAFE. SAFE provides an array of therapeutic activities based on attachment theory, established systemic practice and the known visual processing preferences of people with autism.[31 32] SAFE is best seen as a toolkit with a variety of activities which can be applied to family therapy flexibly. Activities include visual tasks, drawing, modelling, role-play and tracking circular patterns. Sessions are led by family need and the therapists and family work collaboratively, often in a playful way, using family resources, therapist expertise and the tools that SAFE provides. SAFE draws heavily from well-documented active and playful approaches in family therapy practice and literature.[33] Each therapy session included two therapists with a minimum of intermediate family therapy level of qualification and 4 days training in SAFE principles. Between weeks 1 and 16, families allocated to the SAFE intervention attended five 3-hour SAFE therapy sessions. Sessions 1 and 5 were multifamily sessions and took place in a community setting. Sessions 2, 3 and 4 were for individual families and took place in a community venue or the family home. Following completion of the therapy programme, families attended a group follow-up session at 24 weeks postallocation. Trained support workers from local voluntary groups attended this follow-up session and gave the families information about continued support through existing networks.

### Support as usually employed

Families were typically offered a postdiagnosis follow-up appointment with the diagnosing paediatrician.

Parents of children whose symptoms were not severe tended to be directed to local authority parenting classes. In some cases, families were also directed to relevant resources, for example, The National Autism Society, Gateway ASD and the NHS Child and Adolescent Mental Health Services. For families where a member was experiencing depression or anxiety, treatment varied and was not linked to autism-related care. Referral was often through the general practitioner with patients receiving cognitive behavioural therapy as part of the improved access for psychological therapies service and or medication.

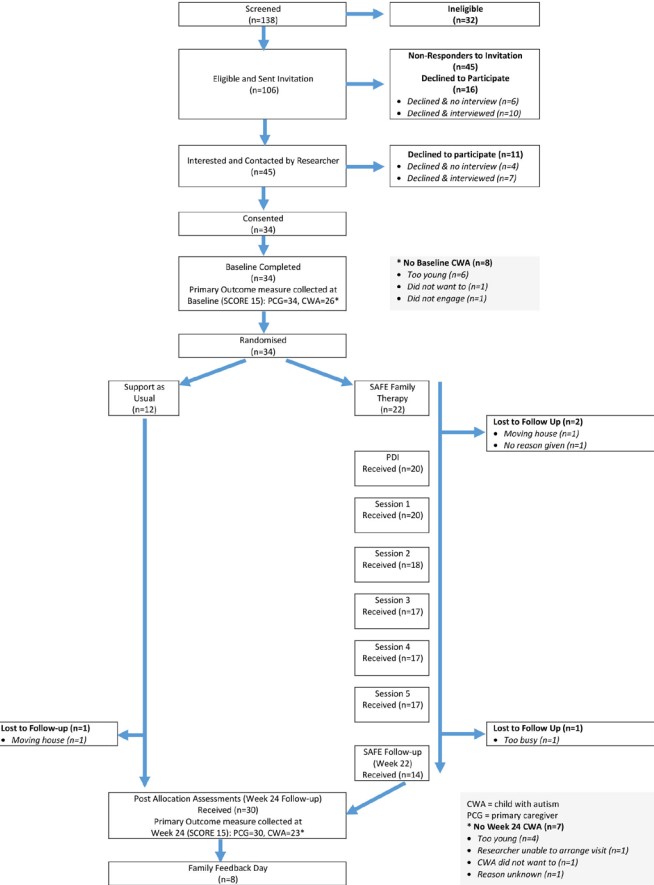

**Figure 1** Participant flow diagram (CONSORT). CONSORT, Consolidated Standards of Reporting Trials; CWA, child with a diagnosis of autism; PCG, primary caregiver; PDI, parent development interview; SAFE, Systemic Autism-related Family Enabling; SCORE, Systemic CORE 15.

## Procedure

Families were recruited in four cohorts of between 6 and 12 families (two cohorts from Plymouth and two from Cornwall). Postbaseline, once sufficient families had been recruited to establish a cohort, families were randomised 2:1 to the SAFE intervention plus Support as Usually Employed (SAFE +SUE) or SUE only. Randomisation was undertaken by a member of the Peninsula Clinical Trials Unit (CTU) programming team by means of a 24-hour web-based system created by the CTU in conjunction with a statistician independent from the trial team, and used random permuted blocks, with stratification by study site. Twenty-two families were allocated to SAFE +SUE and 12 families were allocated to SUE (see figure 1). Clinical staff and research staff collecting outcome data were blinded to allocation. Those in the intervention arm received a preparatory home visit from a therapist.

Outcome measure data were collected at baseline and 24 weeks postrandomisation. Additional data were collected from the intervention participants: post allocation, at each therapy session and at the feedback day (see figure 1).

## Feasibility outcome measures

► Ability to identify, recruit and randomise eligible families.
► Acceptability of proposed outcome measures and follow-up schedule to participants, and whether targets for loss to follow-up were achievable.
► Adherence of therapists and families to the intervention.
► Ability to gather quantitative data on outcomes.
► Appropriateness of RUQ and preference-based instruments for this population.
► Effectiveness and scalability of training arrangements.

## Clinical outcome measures

The proposed primary outcome measure was, the Systemic CORE 15 (SCORE-15).[34] This is a 15-item paper-based survey made up of three subcategories. For each item scores range from 1 to 5. The potential range of overall scores is 15 to 75, with a lower score indicating higher family functioning. Based on available literature,[35] a meaningful change in family function, from preintervention to postintervention, would be solidly indicated by a reduction of 3 on overall scores. The SCORE is the primary measure of family functioning employed in Children and Young People's Improving Access to Psychological Therapies national programme, and is the gold standard for assessing the impact of family therapy on quality of life in the UK.[36] Every able family member was asked to complete the SCORE-15 at baseline and 24 weeks postrandomisation.

The SCORE-15 is particularly well suited as a primary measure for this study as it is a measure for the whole family and, like SAFE, it has a systemic framework based on the premise that relational dynamics are central to the well-being of all family members. In addition, the three dimensions of SAFE are well matched to the needs of this population in that the SCORE-15 is sensitive to change in aspects of mental health and problem solving defined by the three subcategories measuring: family strengths, resilience and coping (Strengths and Adaptability); communication and understanding within the family (Disrupted Communication) and changes in ability to manage problems (Overwhelmed by Difficulties). The SCORE also correlates with measures of parental satisfaction, mental health, strengths and difficulties and life satisfaction.[37]

Scores on the proposed secondary outcome measures, which index changes in child behaviour, child–parent attachment, anxiety and depression:
► Patient Health Questionnaire (PHQ)-Somatic Anxiety Depressive Symptoms completed by parents. This comprises the PHQ-9 measuring depression and the Generalised Anxiety Disorder (GAD-7) measuring anxiety.[38]
► Adapted mutuality subscale of the Coding of Attachment-Related Parenting for use with children with Autism (CARP-A)[39] completed by the whole family. The CARP-A is a validated observational measure of a child with autism's attachment behaviour

towards their carer. The measure was adapted to form a family Lego 'building your house' task.

► The CBCL[40] completed by the Primary Caregiver. This is a 30-item paper-based survey, which detects emotional and behavioural problems.
► The RFQ[40] completed by parents, measures ability to understand own and others' mental states.
► Caregiving Helplessness Questionnaire (CGHQ)[41] completed by the parents. This is a 26-item questionnaire designed to assess aspects of disorganised caregiving.

Scores on proposed generic health economic outcome measures also provided data on the feasibility of cost-effectiveness analysis:

► EuroQol 5 dimensions (EQ-5D), a standardised generic instrument for measuring health outcome.
► Child Health Utility 9 Dimensions (CHU-9D), a paediatric generic preference-based measure of health-related quality of life.
► RUQ, a paper-based questionnaire completed by parent about his/her child's use of healthcare and social resources. The RUQ is designed to identify the NHS and social care resource use for the economic evaluation.

### Proposed qualitative outcomes assessed via focus groups

Proposed qualitative outcomes assessed via focus groups:

► Acceptability of SAFE and the trial process for participants and therapists.
► Reasons for declining and withdrawing from the study.

Additional process measures completed after each SAFE session by therapists and families allocated to SAFE:

► The Client Satisfaction Questionnaire (CSQ-8 completed by families).
► Helpful Aspects of Therapy Questionnaire—assessing what was helpful/unhelpful in each session (HAT completed by families).
► The Training Checklist and Questionnaire monitoring protocol adherence (TCQ completed by therapists).

### Quantitative analysis

Prior to analysis a detailed statistical analysis plan was created and agreed by the research team. For each outcome measure, the relevant scores were calculated and presented descriptively by trial arm.

Where available, published guidelines were used to process, score and summarise the measures including the use of imputation in the event of missing items on a questionnaire. Summary measures were calculated as appropriate, for example, means and SD, medians and ranges, numbers and percentages in categories. The only analysis contrasting the two groups was an interval estimate in the form of a 95% CI for the primary outcome, the SCORE-15.

### Qualitative analysis

Prior to analysis a qualitative analysis plan was agreed. Focus group interviews were conducted for SAFE +SUE and SUE families from each cohort after the 24 weeks outcome data had been collected for that cohort. Focus groups were audio recorded and transcribed verbatim. Consequent qualitative data were managed using proprietary computer-assisted qualitative data analysis software, NVivo V.10, and analysed thematically.[42 43] Summaries of process measures were compiled including quantitative data from Likert scales and thematic analysis of open-ended questions. Subsequently, independent analysis of qualitative data was conducted, including notes and memos, on a subset of the data by two of the authors and consensus reached on overarching themes relating to family experience of SAFE.

## RESULTS

### Recruitment and randomisation

Of 138 families screened, 106 were eligible and 32 ineligible. Forty-five eligible families expressed interest of which 34 were recruited (figure 1). Since randomisation only occurred once a complete cohort (between 6 and 12 families at each site) was recruited, the time between consent (when baseline questionnaires were completed) and randomisation varied. Data on relevant dates were available for 33 families, for whom there was a median (range) of 21 days (2–111) between baseline questionnaire completion and randomisation, with a mean (SD) of 34 (32) days. Nineteen families completed the 24-week assessments for which dates were available. Six of the 19 dates were less than 22 weeks from randomisation due to time constraints, none were more than 26 weeks. All primary caregivers were mothers and there were three single parent families (two allocated to SAFE+SUE and one allocated to SUE). Baseline demographics were similar for children with autism in both groups (See table 1). Baseline constitution of families varied and are shown in table 2.

**Table 1** Summary statistics of baseline and demographic characteristics for children with autism

| | SAFE+SUE (n=22) | SUE (n=12) | Total (n=34) |
|---|---|---|---|
| **Age** | | | |
| Mean (SD) | 9.5 (2.7) | 9.4 (2.4) | 9.5 (2.5) |
| Median (range) | 10 (5–14) | 9 (6–13) | 10 (5–14) |
| **Gender n (%)** | | | |
| Male | 17 (77) | 9 (75) | 26 (76) |
| Female | 5 (22) | 3 (25) | 8 (24) |
| **Ethnicity n (%)** | | | |
| White | 22 (100) | 12 (100) | 34 (100) |

SAFE, Systemic Autism-related Family Enabling; SUE, Support as Usually Employed.

**Table 2** Constitution of families at baseline

| Family members in addition to child with autism | No families SAFE+SUE | No families SUE | No families total |
|---|---|---|---|
| Mother (only) | 2 | 1 | 3 |
| Mother + 1 sibling | 2 | 0 | 2 |
| Mother + 2 siblings | 1 | 0 | 1 |
| Mother + 3 siblings | 2 | 0 | 2 |
| Mother + 4 siblings | 2 | 0 | 2 |
| Mother + father | 0 | 4 | 4 |
| Mother + father + 1 sibling | 6 | 4 | 10 |
| Mother + father + 2 siblings | 3 | 1 | 4 |
| Mother + father + 3 siblings | 1 | 1 | 2 |
| Mother + father + 4 siblings | 0 | 1 | 1 |
| Mother + father + grandmother + grandfather | 1 | 0 | 1 |
| Mother + father + grandmother + grandfather + 1 sibling | 1 | 0 | 1 |
| Mother + 3 siblings +1 nephew | 1 | 0 | 1 |

SAFE, Systemic Autism-related Family Enabling; SUE, Support as Usually Employed.

### Acceptability of outcome measures and follow-up schedule and lost to follow-up

One SUE family was lost to follow-up and three SAFE+SUE families were lost to follow-up (figure 1). Mechanisms were in place to report and record serious adverse events (SAEs) related to mental health from consent until participants completed the follow-up or withdrew from the study. No SAEs were reported.

For the purposes of the study, the core family was taken to be the primary caregiver (all mothers) and children with autism. In one case, parents were joint primary caregivers. Families chose who else would attend sessions and complete data. At baseline, 34 families and 122 family members provided data. At the 24-week assessment, 30 families and 97 family members provided data. Different outcomes are completed by different combinations of family members. The primary caregiver provided data on all measures at baseline, and 22 of the 23 other caregivers (fathers) did likewise. There was a small number of caregivers who did not provide data at 24 weeks for some questionnaires. The completion of the SCORE by the children with autism and siblings was optional. Reasons for non-completion by the child with autism were: child was too young (n=4), the visit could not be arranged (n=1), the child did not want to (n=1) and reason unknown (n=1) (see figure 1).

### Adherence of families to the intervention

Full engagement with the SAFE intervention involved five 3-hour therapy sessions. The first and last of which were multifamily sessions for the parents (children could attend if they wished) between weeks 1 and 16, with a final feedback day at 22 weeks. The primary caregiver was present for these sessions, except for one mother who was absent for session 1, but the father attended. The child with autism was always present at the individual sessions 2–4. Five children were also present at parent sessions 1 and 5.

### Quantitative data on outcomes

Results for the SCORE-15, including expressions of uncertainty (95% CI) for estimates by randomised groups are shown in table 3. In general there is no specific guidance available on how to handle missing data for the outcome measures used. In what follows, a decision was made to use an individual's total score data if there was no more than one missing item, for which the mean of the other items was imputed. If there were totals for subgroups, these were included if there was no more than one missing item within each subgroup, for which the mean of the other items within the subgroup were imputed.

### Feasibility of using economic evaluation instruments for this population

Completeness of health economics data was achieved in intervention and control groups for the primary care based, community based and mental health based services at 100% response rate at baseline dropping to 82% and 83% at 24 weeks in intervention and control groups, respectively. The outpatient-based services had response rates of 68% and 67% at baseline, but completeness improved to 82% and 83% in intervention and control groups, respectively, at 24 weeks.

For EQ-5D-5 level, at baseline the control group had a 100% response rate. At 24 weeks the response rates were 82% and 83% of participants in intervention and control groups, respectively.

As expected CHU-9D responses were largely provided by children with autism (completed by the primary caregiver where necessary). Response rate in the intervention group fell from 77% to 73% for children with autism but remained constant with three (14%) primary caregiver responses at baseline and 24 weeks. In the control group, response rate fell from 83% to 58% in children with autism and 25% to 17% in Primary Caregivers (mothers) from baseline to week 24. Despite indications for increased appropriate service use coupled with reduced cost over time for SAFE+SUE children with autism, the economic evaluation focused only on children with autism, consequently insufficient data was collected for whole family service use and costs.

### The families' experience of the study and the SAFE intervention

Qualitative data on the experience of the study were collected from SAFE+SUE and SUE families in family feedback days for each cohort. Key areas revealed by analysis of the data were (1) the study was not burdensome, (2) that they took part to help others, (3) research

**Table 3** Values for SCORE-15, based on the 'total' computed for each dimension and overall for each person

| | Baseline | | | | 24 weeks post randomisation | | | | Difference* | |
| | SAFE+SUE | | SUE | | SAFE+SUE | | SUE | | | |
| | Mean | SD | Mean | SD | Mean | SD | Mean | SD | 95% CI | N |
|---|---|---|---|---|---|---|---|---|---|---|
| **Strengths and adaptability** | | | | | | | | | | |
| CWA | 11.4 | 3.3 | 12.8 | 3.9 | 12.1 | 4.0 | 12.6 | 4.8 | −3.81 to 3.29 | 21 |
| Primary CG | 10.0 | 2.7 | 9.4 | 2.3 | 10.9 | 3.1 | 10.6 | 2.3 | −1.96 to 1.92 | 30 |
| Other† | 11.0 | 3.6 | 10.2 | 3.0 | 11.1 | 3.9 | 11.2 | 4.6 | −1.59 to 3.87 | 43 |
| Whole family† | 10.8 | 3.3 | 10.5 | 3.2 | 11.2 | 3.7 | 11.3 | 3.9 | −0.89 to 2.07 | 94 |
| **Overwhelmed by difficulties** | | | | | | | | | | |
| CWA | 12.3 | 3.9 | 12.3 | 5.5 | 13.5 | 4.2 | 14.1 | 5.5 | −2.89 to 5.58 | 21 |
| Primary CG | 12.4 | 3.5 | 12.1 | 4.0 | 11.5 | 2.7 | 12.7 | 4.1 | −0.69 to 3.79 | 30 |
| Other† | 12.1 | 3.9 | 12.2 | 3.9 | 11.7 | 3.2 | 13.1 | 4.2 | −0.63 to 3.86 | 43 |
| Whole family† | 12.2 | 3.7 | 12.2 | 4.2 | 12.1 | 3.4 | 13.2 | 4.4 | −0.03 to 2.96 | 94 |
| **Disrupted communication** | | | | | | | | | | |
| CWA | 12.8 | 4.3 | 11.8 | 4.0 | 13.2 | 4.4 | 11.4 | 2.9 | −5.02 to 3.08 | 21 |
| Primary CG | 11.9 | 4.8 | 9.6 | 2.8 | 10.2 | 2.6 | 9.6 | 2.5 | −1.95 to 2.04 | 30 |
| Other† | 12.9 | 4.6 | 12.6 | 4.2 | 12.3 | 3.4 | 11.2 | 3.6 | −2.91 to 1.59 | 43 |
| Whole family† | 12.6 | 4.5 | 11.5 | 3.9 | 11.9 | 3.6 | 10.7 | 3.1 | −2.03 to 0.92 | 94 |
| **Overall score** | | | | | | | | | | |
| CWA | 36.4 | 10.1 | 36.8 | 11.9 | 38.8 | 11.3 | 38.1 | 12.0 | −9.68 to 11.22 | 21 |
| Primary CG | 34.3 | 9.7 | 31.1 | 7.6 | 32.5 | 7.2 | 33.0 | 7.9 | −3.49 to 7.35 | 30 |
| Other† | 36.1 | 10.2 | 34.9 | 9.5 | 35.1 | 7.5 | 35.5 | 10.4 | −3.81 to 7.24 | 43 |
| Whole family† | 35.7 | 9.7 | 34.1 | 9.5 | 35.2 | 8.6 | 35.2 | 9.8 | −2.09 to 5.26 | 94 |

Reduction in values and positive differences in 95% CIs indicate positive change.
*Difference between SUE and SAFE +SUE at 24 weeks follow-up, adjusting for baseline in ANCOVA, based on individuals with complete data at both baseline and 24 weeks assessment; no allowance for hierarchy of data.
†Scores averaged across all relevant family members.
ANCOVA, analysis of covariance; CG, caregiver; CWA, child with a diagnosis of autism; SAFE, Systemic Autism-related Family Enabling; SCORE15, Systemic CORE 15; SUE, Support as Usually Employed.

staff made completing tasks easier and sessions could be organised differently.

The families did not find the completion of CARP-A Lego task and questionnaires onerous:

> didn't mind so much doing like the Lego task I mean like it's something that we like to do anyway … but um the questionnaires didn't have a problem with those either

Families felt that there were too many questionnaires, but the approachable nature of the research staff made the task easier:

> it was all fine … it was A [Research Assistant] that came in did it and she's so friendly and approachable that it made it easy…

Families generally took part in the study for altruistic reasons. They wanted to be part of a study which could potentially increase support postdiagnosis. Feedback from SUE families suggested that this helped them to accept their role in the study:

> I thought actually this could be a way of helping future parents of diagnosed children to get the support and advice that wasn't there when I got the diagnosis

The focus group data also revealed that families felt sessions at home were too distracting, the sessions should be shorter, but more of them and more activities for younger children were needed.

Where families were allocated to the intervention, data on family experience of SAFE were collected via process evaluation measures, the HAT and the CSQ completed by family members after each therapy session and via the family feedback day. Quantitative data from the HAT indicated that SAFE families found the intervention helpful with mean scores for all sessions on the HAT being rated by children and adults as helpful or very helpful. Analysis of all qualitative data from process measures and the feedback day resulted in the following overarching themes: therapist helping reflection, increased understanding, feeling closer, feeling more confident, more able to

reflect and problem solve, improved communication and feeling less isolated.

The multifamily sessions made families feel less alone, more able to talk about difficult issues and less judged.

> Talking to other parents and feeling listened to and supported

> Talking about daily issues and sharing these with the other parents, generally feeling that I am not alone

Families reported being able to step back and reflect, leading to more confidence in tackling family problems.

> Breaking down problem events and looking at causes, outcomes, choices made. Compared a recent to past events and saw progress and maturity in decision making and actions

SAFE therapists were seen as allies and the activities helped with family problems leading to increased understanding and improved mental health:

> Actually being able to hear that I'm doing OK and actually not as bad as what I thought I was, was nice. It was something that helped me with my mental and emotional state

### Effectiveness and scalability of training

The therapists completed the TCQ after each session informing the potential for future scalability and establishing that training methods were effective. The TCQ was a short questionnaire documenting adherence to planned activities in each session, confidence and ease of delivering activities, rating of the effectiveness of activities and any additional resources required. Focus groups and interviews were also conducted with the therapists after completion of the intervention. The therapists felt that SAFE was an inspirational and effective intervention, was non-pathologising and child centred. The training was motivational and thorough. In 82% of sessions lead therapists reported feeling confident, that the sessions were effective and the activities were easy to deliver. The support therapists, who were less experienced, felt confident, that sessions were effective and activities easy to deliver in 63% of sessions. The therapists felt supervision was very helpful. They highlighted that the gap between training and intervention was too long.

### Operational experience moving forward

The chief investigator (CI) kept the study on time and within budget despite multiple challenges. The support and advice of the trial steering committee, the PenCTU and the research team facilitated the development of the skill set of the CI in preparation for a definitive trial.

### DISCUSSION

To our knowledge, this study is unique in evaluating the feasibility and efficacy of a family therapy-based intervention designed specifically for families of children with a diagnosis of autism. The results indicate that a larger trial of SAFE would be feasible. The findings also suggest that offering SAFE for families postdiagnosis is both feasible and potentially efficacious.

Progression to a definitive trial was supported based on predetermined criteria met in the following ways: (1) 34 of the target of 36 families were recruited. Family feedback revealed willingness to undergo randomisation as families felt the study had potential to extend services, (2) 88% of families were retained, (3) There was 100% attendance at appropriate sessions for core family members, (4) For the SCORE-15, complete data were available at both time points and for every dimension for 88% primary caregivers, 62% children with autism and 55% other family members. All primary caregivers retained in the study completed the SCORE-15 at both time points, (5) Collecting health and social care resource utilisation data and health-related quality of life data for children with autism was feasible, (6) Process evaluation and feedback day showed that in 82% of sessions lead therapists felt confident, that sessions were effective and easy to deliver and (7) The qualitative and process data from families indicates tentative potential for efficacy as SAFE was rated as helpful or very helpful in addressing problems. Qualitative analysis revealed the following themes of positive change: therapist helping reflection, increased understanding, feeling closer, feeling more confident, more able to reflect and problem solve, improved communication and feeling less isolated. A common thread across all qualitative data was increased mental well-being among family members receiving SAFE. Although we acknowledge that the current study was not powered to detect a difference, the SCORE-15 showed marked reduction in scores for SAFE+SUE suggesting positive change. A reduction of this magnitude indicates tentative potential proof of efficacy[34 35] (see figure 2).

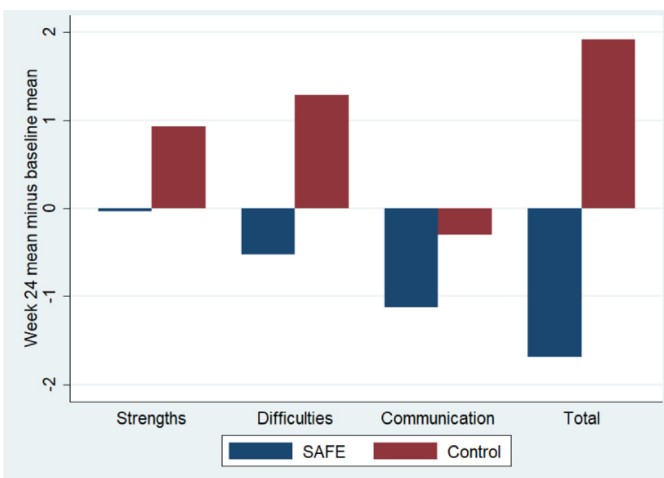

**Figure 2** Mean change in SCORE-15 subcategories for SAFE+SUE (SAFE) and SUE only (control) for all family members, including children with autism, who completed the SCORE-15 at both baseline and at 24 weeks. SAFE, Systemic Autism-related Family Enabling; SCORE-15, Systemic CORE 15; SUE, Support as Usually Employed.

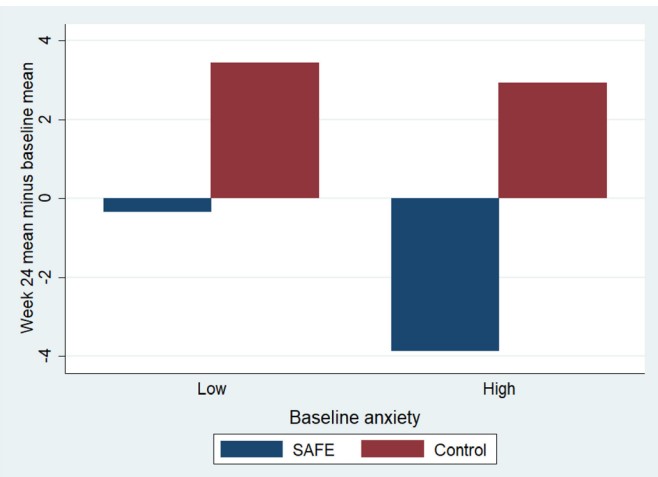

**Figure 3** SCORE-15 total mean change for Caregivers with low and high anxiety (a reduction=positive change; high anxiety >7 on GAD-7). GAD-7, Generalised Anxiety Disorder 7; SAFE, Systemic Autism-related Family Enabling; SCORE-15, Systemic CORE 15.

Additional analysis of the SCORE-15 data revealed that the most marked indications of positive change were experienced by Primary Caregivers with high anxiety as measured by the GAD-7 scale. The relationship between anxiety scores and values on the SCORE-15 is important as SAFE seeks to alleviate mental health issues. An indication that a reduction in anxiety and improved family function can be captured by our proposed measures and that data suggests positive change for those participants with acute difficulties receiving the intervention is therefore promising (see figure 3).

The sample size calculation for a subsequent fully powered randomised controlled trial is based on the SCORE-15 for primary caregivers, which will be the primary outcome in the full trial. The minimal clinically important difference of 3 is based on the literature[35] and an SD of 8 is based on an SD of 7.3 obtained from the feasibility study. To account for a possible group effect in the intervention group, we will use a conservative 0.1 for the intragroup correlation and an assumed mean group size of 6 with variance 2. Employing the conventional two-sided 5% alpha and 90% power, and retaining the 2:1 allocation, gives a requirement for 405 evaluable participants. Using the drop-out experienced in the feasibility study would increase the requirement to 460.

The feasibility study alongside previous research[4–7] strongly suggests that these families are an at-risk group who experience very limited support post diagnosis. One of the primary aims will be to carry out a definitive trial in order to address international[20] and national recommended good practice in relation to children with autism and their families (NICE guidelines).[21 22] These recommend psychosocial interventions, which improve sensitivity, responsiveness and communication within families. They also call for interventions which prevent challenging behaviour by creating care packages which address coexisting mental health problems, and provide support for families. NICE guidelines associated with child mental health also ask for the provision of psychological therapies including family therapy; the need for parents' psychiatric problems to be treated, for children's mental health to improve; and management of developmental conditions to be in parallel with mental health interventions.

NICE research recommendations associated with autism call for randomised controlled trials exploring family interventions that are designed to reduce challenging behaviour, alleviate stress and improve quality of life.[21] Our long-term goals are also informed by the Munroe report,[44] which called for children's services to offer family-centred, therapeutic intervention rather than risk management. Additionally, the current implementation of the improved access to psychological therapies scheme for children[45] promotes proactive attachment-based and family therapy approaches. SAFE has the potential to address this gap by providing a family-based intervention which is sustainable, extends existing services and is beneficial to families.

## LIMITATIONS

Use of the primary measure the SCORE-15 was limited for children below 7 years and an adapted more visual version will be developed so children below 7 years can contribute. The economic evaluation focused solely on the child with a diagnosis. To ensure sufficient data is collected, a subsequent trial should include longer-term service use and costs for all family members. Families felt there were too many measures and repetition. For any subsequent trial the outcome measures would be reduced and adapted. The RFQ replicated information from other measures and the CBCL was lengthy and complex. These measures would not be used in a subsequent trial. We propose the SCORE-15 (measuring perceived change in family functioning) be retained as our primary outcome measure for primary caregiver alongside secondary measures being the adapted SCORE-15 (for other family members including the youngest children), PHQ-SADS (anxiety scale), CGHQ (measuring change in caregiver helplessness) and the CARP-A (measuring mutuality between family members).

Recruitment and organisation of SAFE sessions was challenging. There was large variability in the timescales between baseline and start of intervention. In any subsequent trial waiting could be managed by carrying out baseline measurement once enough families have consented for randomisation to occur. Therapists and families preferred shorter sessions in community venues, predetermined shorter appointments are proposed for any subsequent trial. Challenges anticipated in recruiting for the definitive main trial will be addressed by using a community pathway for recruitment alongside using clinical nurses to aid recruitment in clinical settings. In the current study, the community pathway was introduced

later in the study and did boost recruitment. We will also encourage recruitment and clarity for families by developing a recruitment open day, screening tool and appointment system for SAFE sessions based on a clinic model influenced by potential benefits of family-centred service delivery.[46] In the current study, families of children with autism were not eligible if their children had intellectual disability or needed substantial support. Subsequent refinements of the intervention should address this issue to broaden the application of the intervention. Due to the lack of ethnic diversity in Plymouth and Cornwall our participants were all white. This will need to be addressed in any future trial. Given that prevalence differs between ethnic groups[47–49] and beliefs and support networks may also differ, it is essential for appropriate implementation across the UK, and internationally, that any future trial reflects ethnic diversity. Any definitive trial will, therefore, include centres which have an ethnically diverse client base.

## CONCLUSIONS

To our knowledge, this is the first study of family therapy as an intervention for families of children with a diagnosis of autism. The proposed primary outcome measure, the SCORE-15, indicated potential benefits for the intervention group compared with controls in coping with problems, developing strengths and communicating within families. Qualitative data showed that family members felt the intervention improved understanding, communication, confidence, mental health, reflection, problem solving and sense of isolation. Attendance and completion of outcome measures was acceptable and NHS therapists were effectively skilled after receiving brief SAFE training. This study suggests that SAFE can address current gaps in recommended care postdiagnosis and can and should be subject to a definitive trial. The findings indicate that SAFE can be effectively and confidently delivered by existing NHS staff and has potential to be a feasible and beneficial treatment for these families.

**Acknowledgements** The authors would like to thank the SAFE Family Consultation Group for their input and expertise and all of the participating families from whom we learnt so much. The authors would like to thank James Cook, Mary Hosken, Dr Ben Whalley, Dr Antonieta Medina-Lara and Professor Julian Archer for their help in developing and conducting this study. The authors would also like to give special recognition to the dedication, hard work and sensitivity of Helen Hancocks.

**Contributors** RM, PJV and HH were responsible for the overall development of the study and trial management. RM, HH, RD, CM, PE, AB and TV and the Family Consultation Group were involved in the conception, administration and conduct of the study. PE, JC and AB provided methodological expertise and advice on quantitative analysis, PE and JC provided statistical expertise and analysis. RD was the lead researcher, with the support of JS, on design of the intervention, supervision of therapists and qualitative analysis. TV, with the support of the Family Consultation Group, and CM advised on design, ethics and data collection particularly from the participant perspective. All authors made substantial contributions to drafting, revision and approval of this document.

**Funding** This work was supported by the National Institute for Health Research (NIHR), grant number: PB-PG-0815-20058, and by Autistica, grant number: 7239.

**Competing interests** Contributors are coapplicants or employed research staff on the SAFE project, which receives funds from both NIHR and Autistica. RD holds joint Intellectual Property rights for the SAFE intervention with the University of Plymouth.

**Patient consent for publication** Not required.

**Ethics approval** The trial protocol received ethical approval from the South West-Exeter Research Ethics Committee (Ref: 17/SW/0192).

**Provenance and peer review** Not commissioned; externally peer reviewed.

**Data availability statement** Data are available on reasonable request. Individual participant data that underlie the results reported in this article will be available on request from the lead author and the research Sponsor (University Hospitals Plymouth NHS Trust). Deidentified participant data (including text, tables, figures and online supplemental appendices) will be available from 9 months following publication of the article. Data will be shared with (or access to the data will be provided to) investigators whose proposed use of the data has been approved by the chief investigator and sponsor. In addition, the study protocol will be available on request. For information regarding submitting proposals and accessing data please contact the lead author rebecca.mckenzie@plymouth.ac.uk.

**ORCID iD**
Rebecca McKenzie http://orcid.org/0000-0001-8152-7880

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
