## [Reviewer comments · BMJ Open]

ARTICLE DETAILS

TITLE (PROVISIONAL)	SAFE, a new therapeutic intervention for families of children with autism: a randomised controlled feasibility trial.
AUTHORS	McKenzie, Rebecca; Dallos, Rudi; Stedmon, Jacqui; Hancocks, Helen; Vickery, Patricia; Barton, Andy; Vassallo, Tara; Myhill, Craig; Chynoweth, Jade; Ewings, Paul

VERSION 1 – REVIEW

REVIEWER	Angela Hassiotis University College London
REVIEW RETURNED	01-Apr-2020

GENERAL COMMENTS	Very pleased to see this study. Well conducted and very timely and topical. Also, of interest, in using systemic family therapy. I commend the authors in mentioning challenges and the fact that they've managed the contract to time and costs! The study has already received external funding and I am certain, full external peer review, so here I am only mentioning a few minor points to be taken care of. I have no other concerns.  1. Please add the assessment points in the abstract 2. Related to that, the assessment points are mentioned in the text as baseline and 32 months but in protocol it mentions 24 months follow up. Or indeed another follow up point has been missed? 3. What/how was severity of autism was estimated (1 and 2? not personally aware and possibly other readers too) 4. Not a strong dislike but Support usually deployed? what is wrong with TAU? 5. Please specify the specifics of the autism presentation (I presume social communication) that the systemic therapy is targeting 6. Some more discussion about the treatment even as supplementary material; development of novel treatments needs to be reported in detail 7. The text refers to some places in the future tense, please ensure that all test reports a completed trial. 8. Recruitment was somewhat challenging; less than 50% of those eligible expressed an interest and even fewer took part. How could this issue be mitigated in a UK wide study?
---

REVIEWER	Jessica Amsbary Frank Porter Graham Child Development Institute University of North Carolina at Chapel Hill United States
REVIEW RETURNED	11-May-2020

GENERAL COMMENTS

Thank you for the opportunity to review the manuscript "SAFE, a new therapeutic intervention for families of children with autism: a randomised controlled feasibility trial." I found the content to be useful and important, with the potential to make a significant impact on the field. I do think the manuscript could use a few more details and content clarification to strengthen it prior to publication.

I think stating only the 2 main diagnostic criteria of ASD would be more descriptive in the second sentence (impairments in social communication and the presence of RRBs). I also agree with your participants that you probably had too many secondary measures. The objectives seem a bit overwhelming - I wonder if it would be possible to combine any of them (maybe 4 & 6 for example)? I do not see reference to ethics/Institutional Review Board approval explicitly stated in text.

To better understand the intervention itself (and ensure possible replication) I think more detail is needed in the methodology. The authors mention adherence to the intervention as an outcome that was measured (and that the therapists rate caregiver adherence), but it was not clear to me how therapists were scored on adherence (self checklist)? I think the manuscript would also be strengthened if there were more detail about the what specific "adherence" was measured during each session.

Regarding demographics, were all primary caregivers mothers? If not, I think it would be interesting to know how many were other family members. In addition, for a larger trial, I think it would be very interesting to compare those families with multiple family members involved versus those with just one. For the purposes of this paper, I think it would be helpful to know how many families only had a single caregiver enrolled.

I would like to know a bit more about the qualitative analysis and I think the qualitative data could be presented more coherently. Did consensus coding occur? Any notes/memos kept in coding process? And when presenting the results, I recommend providing an introductory paragraph that presents the main themes (as the authors do in the discussion) and then integrating quotes into text that clearly illustrate and build upon the findings. Some of the quotes don't seem to line up with the text. I think it would be helpful to know how many participants attended focus groups and which data presented came from focus group v. questionnaire...also how many focus groups were held and when?

I would also suggest moving the Families as Partners content to the introduction. With the field focusing more and more on Implementation Science and involving stakeholders in the development and adaptation of interventions, I think this could be a strong driver of the manuscript.

There were a few grammatical errors (for example, on p. 13, line 60 - I think should be "disseminated"?)

Overall, I commend the authors for their work - I think actively involving stakeholders is crucial for successful intervention buy-in and implementation and I think they took important measures to ensure this. This intervention seems to be a promising approach for individuals with autism and their families.

REVIEWER	Kathy Leadbitter University of Manchester, UK
REVIEW RETURNED	01-Jun-2020

GENERAL COMMENTS	Thank you for the opportunity to review this paper, submitted to BMJ Open, entitled 'SAFE, a new therapeutic intervention for families of children with autism: a randomised controlled feasibility trial'. Evidence-based interventions which target systemic family functioning and wellbeing in families of autistic individuals in a clinical and cost effective way are very much needed within the UK context, as well as internationally. I would like to thank the authors for their work in developing, delivering and evaluating the SAFE intervention for this population of families. The paper presents the findings of a feasibility study designed to assess the feasibility of a larger scale trial of this intervention. This study has many positive elements which rightly lead the authors to conclude that a larger trial would be feasible. I have some comments below which I believe would improve the paper. Introduction  • P5 'Explanations for high levels of affective disorders in these families include: stress associated with the condition of autism, genetic factors, and intergenerational family dynamics.' It seems important to include some wider social/societal reasons that cause stress here alongside psychological/biological reasons, such as stigma, lack of social support, constant battles for appropriate provision and support, financial stresses, and so on, e.g. Falk et al 2014, vasilopoulou & Nisbet 2016, Salomone et al 2018, Ludlow, Shelly & Rohleder 2011 • P5 'Previous research demonstrates that experience of trauma and abuse among women is associated with elevated risk of autism developing in their subsequent offspring [13,14]. Hence mothers of children with autism are more likely than the general population to be coping with previous traumatic events.' I am not convinced that we understand enough about these correlational findings and the reasons for these events in a minority of mothers to use them as a justification for the need for this intervention. There is plenty of solid well-understood evidence for raised stress and reduced quality of life in families of autistic children to justify the need for this intervention. Methods  • Exclusion of 'Level 3' children (arguably the most in need of family support) and those with intellectual impairment (around half of all children with autism and presumably making up many of those in Level 2?) – there needs to be a reason/justification for why this intervention is only targeted at/suitable for around half of families of a child with autism. Plus reflection on this within the discussion- what are the implications of this for the wider trial/wider implementation? Would future plans include piloting this intervention with the other 50% of families? Also how was this (level 1 or 2; no ID) operationalised? (did all children have a development assessment to establish their DQ?) • Please spell out 'children with autism/ASD' rather than use CWA in the text. It seems more respectful. Or use 'autistic children' (identity first language is preferred by the autistic community) • P7 'The intervention' - Further detail is required on the intervention for those less familiar with systemic family therapy. I am left with many questions (and readers will want to get a good idea from this paper without being referred elsewhere). How was
---

	the autism-version of the therapy developed and what previous testing has happened, if any? What are the hypothesised mechanisms of change? What exactly does the intervention target? What does 'maladaptive autistic symptoms' mean (a very unclear term)? When it addresses mental health-related difficulties - are these within the child, parent, siblings? Who decides on what the intervention targets are? Are the targets child-focused or parent-focused or both? How much autism-centred practice is built in to increase understanding of and respect for autism-specific coping strategies? What happens within the multi-family sessions and individual sessions to work towards the targets? What do parents take away from the intervention sessions that could lead to lasting change – knowledge, understanding, reassurance, strategies, games to try at home?  • I think calling it a '16-week intervention' is misleading; it is a 5-session intervention with sessions delivered over 16 weeks. • Consort – it would be helpful to have reasons for ineligibility if you have those. • The study has demonstrated ability to identify, recruit and randomise eligible families. However there are questions over the representativeness of these families, particularly because we know that only 45 families opted in from the 106 eligible and invited families. There is a concern that the families that have opted into the study are from certain demographics groups and/or the very families who are already functioning well. There is a lack of ethnic diversity within the sample – and I was pleased to see this addressed in the limitations section. Also, all parents are female. The sample characteristics provided within the table are relatively scant. Is there any other demographic data, e.g. family income, parent occupation, parental mental health (e.g. diagnosed mental health conditions), family structure? It would be good to know more about the sample generally, but also so we can assess how representative the sample is in terms of their demographics and family needs. • 'Data on relevant dates were available for 33 families, for whom there was a median (range) of 21 days (2 to 111) between baseline questionnaire completion and randomisation, with a mean (SD) of 34 (32) days.' There is large variability in the timescales between baseline and start of intervention (and presumably follow-up measurement) and, for some, this is over three months. This is a limitation should be acknowledged within the discussion with reflection on any learning from this for a bigger trial (i.e. carry out baseline measurement once you know you have enough consented parents in a group to randomise?) Results  • No data are provided or discussed around therapy fidelity or adherence of therapists to the intervention manual. It would seem relevant to include this within a feasibility study if it were collected. If not, this should be addressed as a limitation and plans for collecting these data within a larger trial discussed. • 'In more than 70% of sessions therapists reported feeling confident, that the sessions were effective and the activities were easy to deliver.' Please state the exact %. Also this means (probably) that in almost 30% of sessions therapists did not give this positive feedback and I think there needs to more reflection on the implications of this and how this will be addressed Discussion  • P15 'For the SCORE-15, complete data were available at both time points and for every dimension for 88% Primary Caregivers, 62% CWA and 55% other family members.' Further thought needs
--	---

	to be given to the utility of outcomes measures that are completed by only a subset of the sample.  • P15 'The qualitative data from families indicates potential for efficacy as SAFE was rated as helpful or very helpful in addressing problems.' This is an overstatement. There is a huge leap from 'helpfulness' to 'efficacy'. • P15 'Although we acknowledge that the current study was not powered to detect a difference, the SCORE-15 showed marked reduction in scores for SAFE+SUE suggesting positive change. A reduction of this magnitude indicates potential proof of efficacy'. This is overstating the findings and needs to be more nuanced. Firstly, any comment around outcomes or efficacy needs to be very tentative in this small-scale study. Secondly, the confidence intervals in table 2 don't support this conclusion. Thirdly, can we confidently say that a change of 1 or 2 points on a measure would constitute a 'marked reduction'? Linked to that point, thought needs to be given to the clinical significance of any change in scores on the SCORE-15. • The limitations section mainly addresses learning resulting from the study and not limitations of the feasibility study itself (the learning was the point of the study, not a weakness of it) much of this could be embedded within the main results/discussion sections. Figure 2 – this needs to state that it is based on the primary caregiver data.
--	--

REVIEWER	Christine Baulig Institute for Medical Biometry and Epidemiology (IMBE) Department of Human Medicine, Faculty of Health at the University of Witten / Herdecke Alfred Herrhausen-Strasse 50, D-58448 Witten
REVIEW RETURNED	02-Jun-2020

GENERAL COMMENTS	Thank you for this interesting investigation. This review only refers to the regulatory aspects of the presented study. Already in the abstract there is no information on the outcome of the primary endpoint (SCORE 15), the effect size and precision - reported for each group Material and method: How was sample size determined? (there is no information about the sample size calculation), Randomisation: Authors should clearly describe the method for assigning participants to interventions. All informations on randomization are missing (Method used to generate the random allocation sequence, Type of randomisation; details of any restriction (such as blocking and block size). Since the SCORE-15 is unknown to me, I would like to know how many points can be reached and what this means. Results: For the primary outcome (SCORE_15) authors should report trial results as a summary of the outcome in each group (e.g. median, Min/Max), together with the contrast between groups known as the effect size with 95% CI. And I would like a clear statement on the primary endpoint - the table is confusing at first. There is no verbal explanation of what has been found (including the answer to the Question - are these results clinically relevant / statistically significant?). In my view, there are no significant results because the study was underpowered. A sample size calculation is missing.
---

	The discussion lacks of power assessment; assessment of selection bias and responder analysis.
--	--

REVIEWER	Francesco Sera London School of Hygiene and Tropical Medicine
REVIEW RETURNED	05-Jun-2020

GENERAL COMMENTS	This paper presents the results of a randomised controlled feasibility trial of a new therapeutic intervention (SAFE) for families of children with autism. The research question is highly relevant, the paper is well structured and overall is very clear. I'm reviewing the statistical aspects of this paper. Overall the study is well conducted. I have two points that I would discuss with the authors: The authors present descriptive statistics of the main outcome (SCORW-15) at baseline and follow-up by treatment group in Table 2. There are some aspects that are not clear in this table and the coherence with results presented in figure 2: 1) The authors presents only confidence intervals but not point estimates. It is not clear to which point estimates CI refers to. Do they refer to difference at follow up? Are they modelled based, e.g. adjusted by baseline values as reported in the footnote (^). Note that the footnote has no reference in the table. 2) Figure 2 shows (for primary caregiver?) that there is a decline of the score-15 in the intervention group and an increase in the control group. This is consistent with the descriptive statistics presented in Table 2, but the CI limits suggest that there is a positive difference. This is not clear to me. 3) AS the author reported in the footnote (^) for family members possible non-independence should be taken into account in the statistical model. In the discussion section (page 14) the authors noted that the reduction on the score-15 among primary caregiver indicates potential proof of efficacy. I think that it would be also important to give an idea of the sample size required for the full study using the information from the feasibility study.
--

VERSION 1 – AUTHOR RESPONSE

Reviewer 1

1. Please add the assessment points in the abstract. This has now been added (See page 1).
2. Related to that, the assessment points are mentioned in the text as baseline and 32 months but in protocol it mentions 24 months follow up. Or indeed another follow up point has been missed? This has been rectified and assessment at 24 weeks is consistent across both documents (See pages 1 and 5).
3. What/how was severity of autism was estimated (1 and 2? not personally aware and possibly other readers too) This has now been clarified (See page 4).
4. Not a strong dislike but Support usually deployed? what is wrong with TAU? In many cases the families were attending voluntary support groups or local parenting classes rather than receiving autism-related 'treatment'. Hence the term Support as Usually Employed was deemed more appropriate in this case.
5. Please specify the specifics of the autism presentation (I presume social communication) that the

systemic therapy is targeting. This has now been clarified (See page 3).

6. Some more discussion about the treatment even as supplementary material; development of novel treatments needs to be reported in detail. (An appendix has now been added summarising the intervention (See page 4 and Appendix 1).

7. The text refers to some places in the future tense, please ensure that all test reports a completed trial. The document has been checked and now consistently reflects a completed trial throughout.

8. Recruitment was somewhat challenging; less than 50% of those eligible expressed an interest and even fewer took part. How could this issue be mitigated in a UK wide study? Suggestions have now been included for how recruitment might be improved (See page 12).

Reviewer 2

Thank you for the opportunity to review the manuscript "SAFE, a new therapeutic intervention for families of children with autism: a randomised controlled feasibility trial." I found the content to be useful and important, with the potential to make a significant impact on the field. I do think the manuscript could use a few more details and content clarification to strengthen it prior to publication.

I think stating only the 2 main diagnostic criteria of ASD would be more descriptive in the second sentence (impairments in social communication and the presence of RRBs). This has now been done (See page 3). I also agree with your participants that you probably had too many secondary measures. The objectives seem a bit overwhelming - I wonder if it would be possible to combine any of them (maybe 4 & 6 for example)? Thank you for the comments regarding this. We have chosen to respond to each objective in order to clearly show how they were each met. Consequently we have chosen not to combine them.

I do not see reference to ethics/Institutional Review Board approval explicitly stated in text. This has now been added (See page 2).

To better understand the intervention itself (and ensure possible replication) I think more detail is needed in the methodology. An appendix has been added outlining the intervention (See page 5 and Appendix 1). The authors mention adherence to the intervention as an outcome that was measured (and that the therapists rate caregiver adherence), but it was not clear to me how therapists were scored on adherence (self checklist)? I think the manuscript would also be strengthened if there were more detail about the what specific "adherence" was measured during each session. More detail has now been added (See page 11).

Regarding demographics, were all primary caregivers mothers? If not, I think it would be interesting to know how many were other family members. In addition, for a larger trial, I think it would be very interesting to compare those families with multiple family members involved versus those with just one. We note this with thanks for advice in planning the main trial. For the purposes of this paper, I think it would be helpful to know how many families only had a single caregiver enrolled. Information about the gender of primary caregivers and the number of single parent families has now been added (See page 7).

I would like to know a bit more about the qualitative analysis and I think the qualitative data could be presented more coherently. Did consensus coding occur? Any notes/memos kept in coding process? More information has now been provided (See page 7). And when presenting the results, I recommend providing an introductory paragraph that presents the main themes (as the authors do in the discussion) and then integrating quotes into text that clearly illustrate and build upon the findings. Some of the quotes don't seem to line up with the text. I think it would be helpful to know how many participants attended focus groups and which data presented came from focus group v. questionnaire...also how many focus groups were held and when? The qualitative sections have been revised to clarify these issues (See pages 7 and 10).

I would also suggest moving the Families as Partners content to the introduction. With the field focusing more and more on Implementation Science and involving stakeholders in the development and adaptation of interventions, I think this could be a strong driver of the manuscript. The section on Patient and public involvement has been moved to directly follow the introduction.

There were a few grammatical errors (for example, on p. 13, line 60 - I think should be

"disseminated"? The text has been proof read and the particular example has been rectified (See page 4).

Overall, I commend the authors for their work - I think actively involving stakeholders is crucial for successful intervention buy-in and implementation and I think they took important measures to ensure this. This intervention seems to be a promising approach for individuals with autism and their families.

Reviewer 3

Thank you for the opportunity to review this paper, submitted to BMJ Open, entitled 'SAFE, a new therapeutic intervention for families of children with autism: a randomised controlled feasibility trial'. Evidence-based interventions which target systemic family functioning and wellbeing in families of autistic individuals in a clinical and cost effective way are very much needed within the UK context, as well as internationally. I would like to thank the authors for their work in developing, delivering and evaluating the SAFE intervention for this population of families. The paper presents the findings of a feasibility study designed to assess the feasibility of a larger scale trial of this intervention. This study has many positive elements which rightly lead the authors to conclude that a larger trial would be feasible. I have some comments below which I believe would improve the paper.

Introduction

- P5 'Explanations for high levels of affective disorders in these families include: stress associated with the condition of autism, genetic factors, and intergenerational family dynamics.' It seems important to include some wider social/societal reasons that cause stress here alongside psychological/biological reasons, such as stigma, lack of social support, constant battles for appropriate provision and support, financial stresses, and so on, e.g. Falk et al 2014, vasilopoulou & Nisbet 2016, Salomone et al 2018, Ludlow, Shelly & Rohleder 2011. This has now been added based on some of the suggested literature (See page 3).
- P5 'Previous research demonstrates that experience of trauma and abuse among women is associated with elevated risk of autism developing in their subsequent offspring [13,14]. Hence mothers of children with autism are more likely than the general population to be coping with previous traumatic events.' I am not convinced that we understand enough about these correlational findings and the reasons for these events in a minority of mothers to use them as a justification for the need for this intervention. There is plenty of solid well-understood evidence for raised stress and reduced quality of life in families of autistic children to justify the need for this intervention. Our own research (to be published shortly) indicates that this is an important emerging area. We therefore have decided to retain this point whilst strengthening the emphasis on societal issues (See above).

Methods

- Exclusion of 'Level 3' children (arguably the most in need of family support) and those with intellectual impairment (around half of all children with autism and presumably making up many of those in Level 2?) – there needs to be a reason/justification for why this intervention is only targeted at/suitable for around half of families of a child with autism. This has been clarified (See page 4). Plus reflection on this within the discussion- what are the implications of this for the wider trial/wider implementation? Would future plans include piloting this intervention with the other 50% of families? Discussion of this issue is now included in the limitations section of the discussion (See page 12). Also how was this (level 1 or 2; no ID) operationalised? (did all children have a development assessment to establish their DQ?) How and why eligibility was restricted to severity levels 1 and 2 has now been added (See page 4).
- Please spell out 'children with autism/ASD' rather than use CWA in the text. It seems more respectful. Or use 'autistic children' (identity first language is preferred by the autistic community) CWA has been changed to 'child with autism' throughout with the exception of tables and figures where a variety of abbreviations are used.
- P7 'The intervention' - Further detail is required on the intervention for those less familiar with systemic family therapy. I am left with many questions (and readers will want to get a good idea from this paper without being referred elsewhere). How was the autism-version of the therapy developed and what previous testing has happened, if any? What are the hypothesised mechanisms of change? What exactly does the intervention target? What does 'maladaptive autistic symptoms' mean (a very unclear term)? When it addresses mental health-related difficulties - are these within the child, parent, siblings? Who decides on what the intervention targets are? Are the targets child-focused or parent-focused or both? How much autism-centred practice is built in to increase understanding of and respect for autism-specific coping strategies? What happens within the multi-family sessions and

individual sessions to work towards the targets? What do parents take away from the intervention sessions that could lead to lasting change – knowledge, understanding, reassurance, strategies, games to try at home? A summary of the intervention manual has now been included as an appendix. This document addresses many of the questions outlined (See supplementary file - Appendix 1.)

- I think calling it a '16-week intervention' is misleading; it is a 5-session intervention with sessions delivered over 16 weeks. This has been changed as suggested (See page 1).
- Consort – it would be helpful to have reasons for ineligibility if you have those. Please refer to the exclusion criteria on page 5
- The study has demonstrated ability to identify, recruit and randomise eligible families. However there are questions over the representativeness of these families, particularly because we know that only 45 families opted in from the 106 eligible and invited families. There is a concern that the families that have opted into the study are from certain demographics groups and/or the very families who are already functioning well. There is a lack of ethnic diversity within the sample – and I was pleased to see this addressed in the limitations section. Also, all parents are female. All Primary caregivers are female but not all parents please see the table now added on page 8. The sample characteristics provided within the table are relatively scant. Is there any other demographic data, e.g. family income, parent occupation, parental mental health (e.g. diagnosed mental health conditions), family structure? It would be good to know more about the sample generally, but also so we can assess how representative the sample is in terms of their demographics and family needs. An additional table has been added with family constitution at baseline (See page 8).
- 'Data on relevant dates were available for 33 families, for whom there was a median (range) of 21 days (2 to 111) between baseline questionnaire completion and randomisation, with a mean (SD) of 34 (32) days.' There is large variability in the timescales between baseline and start of intervention (and presumably follow-up measurement) and, for some, this is over three months. This is a limitation should be acknowledged within the discussion with reflection on any learning from this for a bigger trial (i.e. carry out baseline measurement once you know you have enough consented parents in a group to randomise?) Thank you for your suggestion this has now been added on page 12.

Results

- No data are provided or discussed around therapy fidelity or adherence of therapists to the intervention manual. It would seem relevant to include this within a feasibility study if it were collected. If not, this should be addressed as a limitation and plans for collecting these data within a larger trial discussed. The collection of data on therapy fidelity adherence via the TCQ is now clarified (See page 11).
- 'In more than 70% of sessions therapists reported feeling confident, that the sessions were effective and the activities were easy to deliver.' Please state the exact %. Also this means (probably) that in almost 30% of sessions therapists did not give this positive feedback and I think there needs to be more reflection on the implications of this and how this will be addressed. The therapists worked in pairs of a lead experienced therapist and a support therapist who were less experienced. This has now been clarified and the exact percentages given for both types of therapist. (See pages 5 and 11).

Discussion

- P15 'For the SCORE-15, complete data were available at both time points and for every dimension for 88% Primary Caregivers, 62% CWA and 55% other family members.' Further thought needs to be given to the utility of outcomes measures that are completed by only a subset of the sample. Thank you for your comments. We are thinking about this carefully for the main trial.
- P15 'The qualitative data from families indicates potential for efficacy as SAFE was rated as helpful or very helpful in addressing problems.' This is an overstatement. There is a huge leap from 'helpfulness' to 'efficacy'. This has been stated as tentative (See page 11).
- P15 'Although we acknowledge that the current study was not powered to detect a difference, the SCORE-15 showed marked reduction in scores for SAFE+SUE suggesting positive change. A reduction of this magnitude indicates potential proof of efficacy'. This is overstating the findings and needs to be more nuanced. Firstly, any comment around outcomes or efficacy needs to be very tentative in this small-scale study. Secondly, the confidence intervals in table 2 don't support this conclusion. Thirdly, can we confidently say that a change of 1 or 2 points on a measure would constitute a 'marked reduction'? Linked to that point, thought needs to be given to the clinical significance of any change in scores on the SCORE-15. We have made this comment more tentative and highlight a reference which gives more information regarding the SCORE-15 (See page 11). We also are giving serious thought to the clinical significance of change in scores for a definitive trial
- The limitations section mainly addresses learning resulting from the study and not limitations of the feasibility study itself (the learning was the point of the study, not a weakness of it) much of this

could be embedded within the main results/discussion sections. We appreciate your comments, but we believe that the learning from the study is appropriate here to avoid repetition, as it responds to limitations discussed.

Figure 2 – this needs to state that it is based on the primary caregiver data. Please see title for the figure provided.

Reviewer 4

Thank you for this interesting investigation.

This review only refers to the regulatory aspects of the presented study.

Already in the abstract there is no information on the outcome of the primary endpoint (SCORE 15), the effect size and precision - reported for each group (Additional information regarding the SCORE-15 has been added on pages 8 and 9. Please also see last comment below)

Material and method:

How was sample size determined? (there is no information about the sample size calculation), We appreciate that one of the key purposes of a feasibility study is to obtain data to inform the sample size calculation for a full-scale trial. Details about the sample size for the RCT now appear in the discussion (See page 12).

Randomisation: Authors should clearly describe the method for assigning participants to interventions. All informations on randomization are missing (Method used to generate the random allocation sequence, Type of randomisation; details of any restriction (such as blocking and block size). This information has now been added (See page 5).

Since the SCORE-15 is unknown to me, I would like to know how many points can be reached and what this means. Details about range of scores and meaningful change on the SCORE-15 have now been added including an additional reference (See page 6).

Results: For the primary outcome (SCORE_15) authors should report trial results as a summary of the outcome in each group (e.g. median, Min/Max), together with the contrast between groups known as the effect size with 95% CI. And I would like a clear statement on the primary endpoint - the table is confusing at first. There is no verbal explanation of what has been found (including the answer to the Question - are these results clinically relevant / statistically significant?). (See points made below) In my view, there are no significant results because the study was underpowered. A sample size calculation is missing. Details about the sample size for the RCT now appear in the discussion (See page 12).

The discussion lacks of power assessment; assessment of selection bias and responder analysis.

This is a feasibility study. The purpose of the study was to assess the feasibility of doing a fully powered definitive randomised controlled trial. As this was a feasibility trial, formal/inferential statistical analysis and hypothesis testing of the outcome measures was not appropriate and thus was not undertaken. The confusion may have arisen as we include a qualitative study and process measures focusing on family experience of the intervention which goes beyond what is usually expected in a feasibility study. The only analysis contrasting the two groups was an interval estimate in the form of a 95% confidence interval (CI) for the primary outcome, the SCORE-15. the planned statistical methods used in the primary report of the trial results were agreed following the SAP Guideline¹, CONSORT extension for Pilot and Feasibility Studies² and also taking cognisance of the CONSORT extensions for reporting patient-reported outcomes³. We appreciate that one of the key purposes of a feasibility study is to obtain data to inform the sample size calculation for a full-scale trial. This will be conducted prior to a definitive RCT.

1. Gamble, C., et al., Guidelines for the Content of Statistical Analysis Plans in Clinical Trials. JAMA, 2017. **318**(23): p. 2337-2343.
2. Eldridge, S.M., et al., CONSORT 2010 statement: extension to randomised pilot and feasibility trials. BMJ, 2016. **355**: p. i5239.
3. Calvert, M., et al., Reporting of patient-reported outcomes in randomized trials: the CONSORT PRO extension. JAMA, 2013. **309**(8): p. 814-22.

Reviewer 5

This paper presents the results of a randomised controlled feasibility trial of a new therapeutic intervention (SAFE) for families of children with autism.

The research question is highly relevant, the paper is well structured and overall is very clear. I'm reviewing the statistical aspects of this paper. Overall the study is well conducted. I have two points that I would discuss with the authors:

The authors present descriptive statistics of the main outcome (SCORW-15) at baseline and follow-up by treatment group in Table 2.

There are some aspects that are not clear in this table and the coherence with results presented in figure 2: More information regarding the table has been added for clarification (See pages 8 and 9).

1) The authors presents only confidence intervals but not point estimates. It is not clear to which point estimates CI refers to. Do they refer to difference at follow up? Are they modelled based, e.g. adjusted by baseline values as reported in the footnote (^). Note that the footnote has no reference in the table. Yes they refer to difference between SUE and SAFE + SUE at follow up, adjusting for baseline. The footnote and icon (^) have been clarified See pages 8 and 9). Point estimates have not been included due to the uncertainty associated with a small sample so 95% confidence intervals for the differences have been included only.

2) Figure 2 shows (for primary caregiver?) The figure shows mean change in SCORE-15 subcategories for SAFE+SUE (SAFE) and SUE only (Control) for all family members including children with autism. The title of the figure has been changed to reflect this. that that there is a decline of the score-15 in the intervention group and an increase in the control group. This is consistent with the descriptive statistics presented in Table 2, but the CI limits suggest that there is a positive difference. This is not clear to me. For the SCORE-15 table (was Table 2 but is now Table 3), the 95% CI for the difference is the SUE group subtracted by the SAFE+SUE group at 24 weeks post randomisation, adjusted for baseline. As a reduction in the SCORE-15 is considered a better outcome, this means for the SAFE intervention to be successful we would look for a positive difference in the 95% CI. This agrees with Figure 2 and the rest of Table 3.

3) AS the author reported in the footnote (^) for family members possible non-independence should be taken into account in the statistical model. Family member type (i.e. child with autism/ primary caregiver/ other caregiver/ other family member) was not adjusted for in the model. Instead, separate 95% CIs for the difference (i.e. SUE group subtracted by SAFE+SUE group) at 24 weeks post randomisation, adjusted for baseline for each family member type were presented in the SCORE-15 table (i.e. now Table 3).

In the discussion section (page 14) the authors noted that the reduction on the score-15 among primary caregiver indicates potential proof of efficacy. I think that it would be also important to give an idea of the sample size required for the full study using the information from the feasibility study. Discussion of sample size for the main trial is now included (See page 12).

VERSION 2 – REVIEW

REVIEWER	prof Angela Hassiotis UCL Division of Psychiatry, University College London, UK
REVIEW RETURNED	13-Aug-2020

GENERAL COMMENTS	This is a feasibility study with appropriate objectives and the authors have already addressed adequately many of the points raised by reviewers. That includes many detailed that were originally omitted. What I would like to see more of is a discussion of how the full sample of 460 children might be recruited in a study which recruits about 30% of the eligible participants. There is a discrepancy between the section that summarises the findings per objective (e.g. recruited 34 of 36 participants) but in the limitations, there is discussion about challenges. I accept that the authors can not predict the future, but as so much discussion is about the definitive
--

	trial, it would be important to consider mitigation strategies. These need to be added in summary in the present manuscript.
--	--

REVIEWER	Jessica Amsbary University of North Carolina at Chapel Hill (United States)
REVIEW RETURNED	24-Aug-2020

GENERAL COMMENTS	Thank you for the opportunity to review the manuscript, “SAFE, a new therapeutic intervention for families of children with autism: a randomised controlled feasibility trial”. I think this study provides useful and relevant information, but I think it could use some further refinement and additional detail surrounding parent and family perspectives as they relate to SAFE would strengthen the manuscript. I think simplifying and clearly defining objectives would additionally strengthen the manuscript. Some specific recommendations are described below. In the abstract results, I think it should also be mentioned that families found the intervention to be acceptable in addition to the study. As a reader, I am more interested in families’ perspectives as they relate to the intervention, but I recognize the importance of accepting the study design and procedures in a feasibility study as well. In the first paragraph, consider re-wording “high autistic traits” when describing family members. Maybe use broader autism phenotype? This reads a little harsh to me. When the authors describe – “Families of children with autism exhibit psychological comorbidities alongside autism...” it is unclear whether the authors suggest family members are also diagnosed with autism and comorbidities OR if family members have psychological needs in addition to the child having autism? Please reword to clarify. I am not sure the authors should claim that mothers of children with autism are more likely than the general population to be coping with previous traumatic events if trauma is associated with a child being diagnosed with autism. Perhaps add “It is possible that...” or specify to whom in the general population they are referring. What about individuals who have experienced trauma who aren’t mothers at all? I think the authors’ argument could be strengthened if they were to pull in a bit more about benefits of providing family-centered services – using a family systems approach to build parent and family capacity, and empower families. I would direct the authors to Trivette et al., 2010 to begin exploring more details around family centered service delivery. When the authors are providing a description of SAFE and how the intervention addresses autism related needs should be consistent (p. 6) – so there should be a challenges before problem-solving (or something similar). I would also caution against using the terms “treating this condition” as a goal, maybe focus more on addressing needs/minimizing challenges as in the previous sentence. I think it might be helpful to include some of the survey data such as what were the articulated needs of the families? Could the authors provide some examples? What does it mean when you say research emerged from difficulties articulated by families and worked with them to develop solutions? Please provide more detail around this. When describing objectives, objective 5 “Adapt existing resource use questionnaire and assess the feasibility of preference based instruments for this population to facilitate a future economic
--

	evaluation” is confusing. Could more description be provided? Overall, the objectives are a bit overwhelming and the authors may consider combining and/or honing in on a couple of the objectives (perhaps child and family perceived outcomes)? How was family adherence measured? Is it just attendance? Or were there additional fidelity measures in place? If the authors are using quotes about assessments (Lego experience), a little more detail on the measures would be helpful in understanding what the families are talking about. I think additional information on the qualitative data analysis process is needed (beyond saying thematic coding). Were there any member check measures? Any examples to support/describe the consensus coding? More detailed analysis procedures could enhance the relevance of the paper. The determined qualitative themes seem to overlap and are not all described in detail. It should be clearly stated and differentiated between when families are talking about “the study” (or assessments) versus intervention components or benefits. When describing qualitative findings, there is a lot of information collapsed into single ideas. For example, parents feeling less alone, more able to talk about issues, and less judged are three different ideas and all could be expanded. Similarly, reflecting and problem solving are linked together but very different ideas. I think that parents are also sharing they considered therapist as allies, which goes well beyond therapists’ helping families reflect – this is a powerful point for empowering parents and families – consider expanding upon ways the therapists were allies. I am not sure that the TCQ actually measures scalability? I think the only way to really measure scalability is to scale up the intervention using an implementation science/improvement science framework. I think the authors could say the measure informs the potential for future scalability, but I would caution against the claim that it measures scalability. It sounds like the measure might have a bit to do with learning to do the intervention (so could maybe claim that the training methods were effective)? I also think the authors need to include some more interpretation and relevance of their qualitative findings in the discussion (such as the therapists as allies) and consider expanding upon the need for and benefits in including families in the development and implementation of interventions. The findings are not thoroughly interpreted or tied back into the literature. In all, I think the manuscript would be more impactful if the authors were to focus on SCORE changes and further describe and interpret qualitative data (analysis and findings) from families. Focusing on families’ perceived benefits of the intervention and perceived positive changes would provide a strong argument for testing the efficacy of the intervention in a larger scale RCT. I also think the authors should consider adding more information on how including families/employing a family centered approach/creating collaborative partnerships with families leads to improved outcomes.
--	--

REVIEWER	Francesco Sera London School of hygiene and Tropical Medicine
REVIEW RETURNED	24-Aug-2020

GENERAL COMMENTS	The authors answered positively to the points I made in my previous review. My only concern is the link between Table 3 and Figure 2. Perhaps I miss something but In table 3 in the SAFE+SUE group
--

	the overall score-15 decline by 0.5 during the follow up period while in Figure 2 the decline is neraly 2. Similarly the increase in Figure 2 in the SUE group doesn't correspond to the avearge values shown in Table 3.
--	---

VERSION 2 – AUTHOR RESPONSE

Reviewer: 1

This is a feasibility study with appropriate objectives and the authors have already addressed adequately many of the points raised by reviewers. That includes many detailed that were originally omitted.

What I would like to see more of is a discussion of how the full sample of 460 children might be recruited in a study which recruits about 30% of the eligible participants. There is a discrepancy between the section that summarises the findings per objective (e.g. recruited 34 of 36 participants) but in the limitations, there is discussion about challenges. I accept that the authors can not predict the future, but as so much discussion is about the definitive trial, it would be important to consider mitigation strategies. These need to be added in summary in the present manuscript. This has now been added in the limitations section (Page 15).

Reviewer: 2

Reviewer Name: Jessica Amsbary

Thank you for the opportunity to review the manuscript, “SAFE, a new therapeutic intervention for families of children with autism: a randomised controlled feasibility trial”. I think this study provides useful and relevant information, but I think it could use some further refinement and additional detail surrounding parent and family perspectives as they relate to SAFE would strengthen the manuscript. I think simplifying and clearly defining objectives would additionally strengthen the manuscript. Thank you for the comments regarding this. We have chosen to respond to each objective in order to clearly show how they were each met. Consequently we have chosen not to combine them.

Some specific recommendations are described below.

In the abstract results, I think it should also be mentioned that families found the intervention to be acceptable in addition to the study This appears in the abstract on page 2. As a reader, I am more interested in families’ perspectives as they relate to the intervention, but I recognize the importance of accepting the study design and procedures in a feasibility study as well.

In the first paragraph, consider re-wording “high autistic traits” when describing family members. Maybe use broader autism phenotype? This has now been changed based on your suggestion (see page 3). This reads a little harsh to me.

When the authors describe – “Families of children with autism exhibit psychological comorbidities alongside autism...” it is unclear whether the authors suggest family members are also diagnosed with autism and comorbidities OR if family members have psychological needs in addition to the child having autism? Please reword to clarify. This has been reworded (see page 3).

I am not sure the authors should claim that mothers of children with autism are more likely than the general population to be coping with previous traumatic events if trauma is associated with a child being diagnosed with autism. Perhaps add “It is possible that...” or specify to whom in the general population they are referring. What about individuals who have experienced trauma who aren’t mothers at all? The phrase you suggest has been added (see page 3)

I think the authors’ argument could be strengthened if they were to pull in a bit more about benefits of providing family-centered services – using a family systems approach to build parent and family capacity, and empower families. I would direct the authors to Trivette et al., 2010 to begin exploring more details around family centered service delivery. Thank you for these thoughts and the reference we have included this in our discussion of next steps responding to limitations (see pages 15 and 17).

When the authors are providing a description of SAFE and how the intervention addresses autism related needs should be consistent (p. 6) – so there should be a challenges before problem-solving (or something similar). I would also caution against using the terms “treating this condition” as a goal, maybe focus more on addressing needs/minimizing challenges as in the previous sentence. This has now been addressed and the wording changed (see page 3).

I think it might be helpful to include some of the survey data such as what were the articulated needs of the families? Could the authors provide some examples? What does it mean when you say research emerged from difficulties articulated by families and worked with them to develop solutions? Please provide more detail around this. It is difficult to add too much detail here due to limits of the word count. We have, however included the two key themes that emerged from the survey (see page 3).

When describing objectives, objective 5 “Adapt existing resource use questionnaire and assess the feasibility of preference based instruments for this population to facilitate a future economic evaluation” is confusing. Could more description be provided? Overall, the objectives are a bit overwhelming and the authors may consider combining and/or honing in on a couple of the objectives (perhaps child and family perceived outcomes)? Given that the study is complete we cannot change the wording of the objectives at this stage. We recognise and appreciate the points the reviewer is making about combining objectives to improve the flow of the paper as raised above and in the previous comments from this reviewer, but, as stated previously we feel that this paper is a very important document in supporting our progression to a main trial as a means of demonstrating that we met each individual objective and that this is more important than improving the flow of the written word.

How was family adherence measured? Is it just attendance? Or were there additional fidelity measures in place? This is a very interesting point. Each family is free to engage with the activities based on how they interpret them and what they find useful. This kind of flexibility is difficult to marry with words like adherence and we wonder if this is what the reviewer is alluding to. Since each family brings different ways of being, successes and problems to a session it is difficult to generalise what adherence might look like. Consequently as an we measured attendance and asked for them to comment on the helpful/unhelpful aspects of activities in each session through the HAT (see page 7). If the authors are using quotes about assessments (Lego experience), a little more detail on the measures would be helpful in understanding what the families are talking about. More information appears on page 6.

I think additional information on the qualitative data analysis process is needed (beyond saying thematic coding). Were there any member check measures? Any examples to support/describe the consensus coding? More detailed analysis procedures could enhance the relevance of the paper. More information about qualitative analysis appears on page 7. The determined qualitative themes seem to overlap and are not all described in detail. It should be clearly stated and differentiated between when families are talking about “the study” (or assessments) versus intervention components or benefits.

When describing qualitative findings, there is a lot of information collapsed into single ideas. For example, parents feeling less alone, more able to talk about issues, and less judged are three different ideas and all could be expanded. Similarly, reflecting and problem solving are linked together but very different ideas. I think that parents are also sharing they considered therapist as allies, which goes well beyond therapists’ helping families reflect – this is a powerful point for empowering parents and families – consider expanding upon ways the therapists were allies. The reviewer raises some important points here, which may have wider relevance to the relatively small role that qualitative studies play in trials of this kind. The main focus of this study is to show that a larger trial is feasible rather than the qualitative experience of the families and there is also of course, a word count constraint. Nevertheless, we share the reviewer’s thoughts and consequently we secured additional funding for a qualitative study of the family experience of the intervention. Whilst we cannot repeat the findings of this parallel funded study here we direct the reviewer to our sister publication which describes qualitative procedures and findings in much more depth and will be published in

Contemporary Family Therapy shortly.

I am not sure that the TCQ actually measures scalability? I think the only way to really measure scalability is to scale up the intervention using an implementation science/improvement science framework. I think the authors could say the measure informs the potential for future scalability, but I would caution against the claim that it measures scalability. It sounds like the measure might have a bit to do with learning to do the intervention (so could maybe claim that the training methods were effective)? These points have been included and the wording changed (see page 12).

I also think the authors need to include some more interpretation and relevance of their qualitative findings in the discussion (such as the therapists as allies) and consider expanding upon the need for and benefits in including families in the development and implementation of interventions. Please see pages 3 and 4. The findings are not thoroughly interpreted or tied back into the literature.

In all, I think the manuscript would be more impactful if the authors were to focus on SCORE changes and further describe and interpret qualitative data (analysis and findings) from families. Focusing on families' perceived benefits of the intervention and perceived positive changes would provide a strong argument for testing the efficacy of the intervention in a larger scale RCT. Thank you for these points as stated above to avoid repetition in papers for different funding streams we direct the author to our qualitative paper to be published shortly in Contemporary Family Therapy. I also think the authors should consider adding more information on how including families/employing a family centered approach/creating collaborative partnerships with families leads to improved outcomes. We are grateful for the opportunity to look further into family centred services and will keep this in mind in our next steps (see page 15). In addition, we will continue to work in partnership with families and (as is the case in this paper) include family members as co-authors, research designers, advisors and consultants (see pages 3 and 4).

Reviewer: 5

Reviewer Name: Francesco Sera

The authors answered positively to the points I made in my previous review.

My only concern is the link between Table 3 and Figure 2. Perhaps I miss something but In table 3 in the SAFE+SUE group the overall score-15 decline by 0.5 during the follow up period while in Figure 2 the decline is nearly 2. Similarly the increase in Figure 2 in the SUE group doesn't correspond to the average values shown in Table 3. – The scores should not be compared in this way. Participants were only included in Figure 2 if they filled in the SCORE-15 questionnaire at both baseline and week 24 as a difference between the two time points had to be calculated. In table 3, this was based on 118 participants at baseline and 95 at week 24 whereas for Figure 2, this was based on 94 participants (i.e. the number that had complete data for the SCORE-15 at both time points). Additional information has been added clarifying this in the caption for Figure 2 (page 13).

VERSION 3 – REVIEW

REVIEWER	Jessica Amsbary University of North Carolina at Chapel Hill, NC, United States
REVIEW RETURNED	16-Dec-2020

GENERAL COMMENTS	Thank you for the opportunity to review the revised manuscript: SAFE, a new therapeutic intervention for families of children with autism: a randomized controlled feasibility trial. The authors have made many edits and adaptations to the manuscript that addressed a number of my previous concerns and the manuscript provides useful relevant information related to engaging stakeholders in intervention development and implementation.
---

	I do have a few more questions and suggestions. In the Strengths and Limitations section, the authors may want to briefly describe the gap in available research data that the study addressed. In the introduction, a brief description of the broader autism phenotype would also be helpful...might need to add a sentence. In the second paragraph, intergenerational family dynamics could use some clarification. The following sentence seems a little long and the authors may want to consider breaking it up: "The current study should be placed in the context of international calls for improved services and care for families of children with autism at country level [20], alongside National Institute of Health and Care excellence (NICE) guidelines and recommendations [21,22], as well as developments regarding children's service provision proposed by the Munroe Report [23,24]". I personally am unfamiliar with the Munroe report so perhaps including what it says about children's service provision would be useful...I see this info described in the discussion, but it may make more sense in the intro (and then readdress without the additional info in the discussion). In Patient and Public Involvement, it is unclear to me whom the representative for interested families was. Was the representative hired as a RA to be the representative or was she already representing the families in some way before becoming the RA? The following sentence may benefit from being broken up into two sentences and/or clarified as it is a bit confusing: "We feel that this way of working made a substantial impact on the outcomes for the families receiving the intervention and the fact that the feedback from families was overwhelmingly positive". In The Intervention, the following sentence might make more sense if it were broken up: Sessions are led by family need and the therapists and family work collaboratively, often in a playful way, using family resources, therapist expertise and the tools that SAFE provides". An example of what a "tool" is would also be helpful here. One final thought – it might be worthwhile to include the need for adaptations to the model for individuals with more severe symptoms of autism and how this will be addressed in the future. Overall, I commend the authors on this work and I look forward to reading more about this family-centered intervention in the future.
--	---